# Generation of a genetically double-attenuated *Plasmodium berghei* parasite that fully arrests growth during late liver stage development

Melanie Schmid[1,2☯], Raphael Beyeler[1,2☯], Reto Caldelari[1], Ruth Rehmann[1], Volker Heussler[1], Magali Roques[1]*

1 Institute of Cell Biology, University of Bern, Bern, Switzerland, 2 Graduate School for Cellular and Biomedical Sciences, University of Bern, Bern, Switzerland

☯ These authors contributed equally to this work.
* magali.roques@unibe.ch

**Data Availability Statement:** All relevant data are within the paper and its Supporting information files.

## Abstract

Malaria caused by *Plasmodium* parasites remains a large health burden. One approach to combat this disease involves vaccinating individuals with whole sporozoites that have been genetically modified to arrest their development at a specific stage in the liver by targeted gene deletion, resulting in a genetically attenuated parasite (GAP). Through a comprehensive phenotyping screen, we identified the *hscb* gene, encoding a putative iron-sulfur protein assembly chaperone, as crucial for liver stage development, making it a suitable candidate gene for GAP generation. Parasites lacking *Plasmodium berghei* HscB (*Pb*HscB) exhibited normal sporozoite production in mosquitoes, but their liver stage development was severely impaired, characterized by slow growth and delayed expression of merozoite surface protein 1 (MSP1). *In vivo* experiments demonstrated that *Pb*HscB-deficient parasites exhibited a delay in prepatency of 2–4 days, emphasizing the significance of *Pb*HscB for exo-erythrocytic development. Although knockout of *Pb*HscB alone allowed breakthrough infections, it is a potent candidate for a dual gene deletion strategy. PlasMei2, an RNA-binding protein, was previously found to be crucial for the completion of liver stage development. We generated a *Pb*HscB-*Pb*Mei2-double attenuated parasite line, serving as a late liver stage-arresting replication-competent (LARC) GAP, providing a solid block of liver-to-blood stage transition.

## Introduction

Malaria continues to impose a significant health burden, caused by *Plasmodium* parasites. Prior to the symptomatic blood stage, the parasite undergoes extensive replication within hepatocyte. Female *Anopheles* mosquitoes inject approximately 500–1000 *Plasmodium* sporozoites into the human host's skin [1, 2]. The sporozoites transmigrate through epidermal cells and endothelia to reach a blood vessel where they are passively transported by the bloodstream until reaching a liver sinusoid [3]. There, they attach to the endothelium and penetrate endothelial cells to enter the liver tissue [4]. After traversing several hepatocytes, the sporozoite

**Funding:** This research was supported by the Swiss National Science Foundation (SNSF) (grant number 310030_182465) and the Multidisciplinary Center for Infectious Diseases (MCID) grant MA-09 to Volker Heussler. The funders had no role in study design, data collection and analysis, decision to publish, or preparation of the manuscript.

**Competing interests:** The authors have declared that no competing interests exist.

invades its final host cell, initiating a round of asexual replication resulting in tens of thousands of daughter merozoites [5]. These merozoites exit the liver tissue encapsulated in vesicles known as merosomes [6]. Upon merosome rupture, the merozoites rapidly infect erythrocytes, initiating the asexual blood stage development.

Vaccine development against the exo-erythrocytic stage of the parasite has been a long-standing focus due to the low parasite numbers and absence of symptoms during this crucial stage. In 2021, the World Health Organization (WHO) approved the widespread use of the RTS,S/AS01 malaria vaccine, a virus-like particle (VLP) vaccine based on the circumsporozoite protein (CSP) [7]. This vaccine has shown potential to reduce severe malaria risk in young children, with an efficacy under 20% [8]. Additionally, Ghana has recently approved the use of another VLP vaccine, R21, based on CSP, with potential efficacy exceeding 75% [9, 10]. However, VLP vaccines may not provide optimal long-term protection [11]. An alternative strategy for malaria protection is immunization with attenuated whole sporozoites, which can elicit a broader immune response mediated by CD8+ T cells, essential for protective immunity [12–15]. There are three main approaches for sporozoite attenuation: irradiation, chemical attenuation, and genetic attenuation. Radiation-attenuated sporozoites (RAS) have been extensively studied but are challenging to implement due to the dose-dependent effect, where excessive radiation completely arrests sporozoites, resulting in weak protection [16–18]. Chemically attenuated sporozoites show strong protection against subsequent infections but are difficult to administer in the field without medical supervision [19–23]. Ideally, the parasite should effectively be controlled and arrested at a specific time point during the liver stage [18, 24, 25]. Genetically attenuated parasites (GAPs) have been used to achieve this controlled arrest of development. Initially, first-generation GAPs relied on gene deletions that halted the parasite's progression at an early stage after invading hepatocytes. It has been demonstrated that Upregulated in Infective Sporozoites (UIS) 3 and 4 are essential for early liver stage development in *P. berghei*, but homologues of these genes have not been identified in *P. falciparum* yet [26, 27]. Other potential candidates for early arrest are two *Plasmodium*-specific 6-Cys proteins known as P36 and P52, whose deletion leads to early arrest after sporozoite transformation [28, 29]. Despite promising results in mice, a phase I/IIa clinical trial using *P. falciparum* GAPs resulted in breakthrough infections [30] underscoring the importance of developing a safe and fully arresting GAP. Nevertheless, the shortcoming of early liver stage-arresting replication-deficient (EARD) parasites results in a limited immune response against *Plasmodium*, and to circumvent this, nowadays research tends to target late parasite development and asexual blood stage. A promising candidate for a late liver stage-arresting replication-competent (LARC) GAP was discovered through a screening of RNA-binding proteins in *P. yoelii* [31]. This candidate, a Mei2-like RNA-binding protein named PlasMei2, exhibited transcription during the liver stage in *P. yoelii, P. berghei and P. falciparum* [31–33] and during asexual blood stage in *P. berghei* and *P. falciparum* GAP [32, 33]. Deletion of *PlasMei2* led to a complete arrest at a very late stage, just prior to merozoite formation. Immunization of BALB/c mice with PlasMei2-KO *P. yoelli* parasites resulted in significant protection without breakthrough infections [31]. In a humanized mouse model of *P. falciparum*, PlasMei2 was also found to be crucial for liver stage completion, further emphasizing its potential as an excellent candidate for a LARC GAP [32, 33].

However, the *P. yoelli*-BALB/c and *P. falciparum*-humanized mouse models have limitations for vaccination studies [34, 35], in that they might obscure the risk of breakthrough infections. To mitigate this risk, recent studies have reported the dual deletion of *PlasMei2* and *linup* genes in *P. falciparum* and *P. yoelii*, referred to as *Pf*LARC2 and *Py*LARC2 respectively [36]. These studies have shown no breakthrough blood stage infection for *Pf*LARC2 in humanized mice and in the *Py*LARC2 in rodent malaria model, demonstrating the promising

generation of live double-attenuated parasites for further vaccination efforts [36]. Since it is not known whether PlasMei2 and LINUP have a redundant function that could facilitate breakthrough infections, it is crucial to identify other proteins having different functions.

In our approach we identified a suitable gene for a second deletion to complement the Plas-Mei2-KO aiming to engineer a safe and effective parasite vaccine strain.

Previously, we conducted a high-throughput phenotyping screen to identify novel genes that play a crucial role in liver stage development [37]. Using barcode technology, we examined over 1'300 genes in the rodent malaria parasite *Plasmodium berghei*. Our investigation led to the identification of 185 genes that are significantly involved in liver stage development, many of which are associated with key metabolic pathways such as fatty acid synthesis (FASII) and heme synthesis [37], as well as genes involved in gene regulation and protein modification. Among the genes we discovered, several molecular chaperones stood out, particularly the putative mitochondrial iron-sulfur (Fe-S) protein assembly chaperone HscB (PBANKA_0821000). Within this study, parasites lacking the *hscb* gene exhibited a 19-fold reduction in barcode abundance at the salivary gland sporozoite (SG) to the blood of sporozoite-injected mouse (B2) transition demonstrating a strong growth delay phenotype during parasite liver stage development [37]. HscB belongs to the Hsp40 protein family, known as heat shock cognate B chaperones, and it operates in conjunction with an Hsp70 family member, HscA, for the assembly of Fe-S proteins [38, 39]. Fe-S proteins possess an iron sulfur cluster (ISC) as a cofactor, enabling efficient electron transport and involvement in various cellular redox reactions. Examples of Fe-S proteins include aconitase of the tricarboxylic acid (TCA) cycle and succinate dehydrogenase of the respiratory chain [40, 41]. In most eukaryotic cells, the biogenesis of ISCs and Fe-S proteins occurs within the mitochondria via the ISC pathway [41, 42]. *Plasmodium* parasites, however, possess an additional ISC biogenesis pathway in the apicoplast, a vestigial plastid resembling the chloroplast [43, 44]. This pathway is known as the sulfur mobilization (SUF) pathway and has been shown to be essential for asexual blood stage development [45]. Conditional knockout experiments targeting SufS, the first enzyme in the SUF pathway during oocyst development, resulted in impaired sporozoite formation, underscoring the significance of the SUF pathway in sporozoite development [46]. Since the deletion of *hscB* in *P. berghei* parasites did not exhibit significant impairment during the mosquito stage development [37], we wanted to study the effect of *PbhscB*-deletion during liver stage development and generated a clonal *Pb*HscB-KO parasite line. We also created a GFP-tagged line (*Pb*HscB-GFP) to study the localization of the protein. Additionally, we generated a *Pb*HscB-*Pb*Mei2-double KO (*Pb*HscB-*Pb*Mei2-dKO) parasite line, which holds potential as a safe candidate for a late LARC GAP.

## Results

### *Pb*HscB-GFP localizes to the mitochondrion in the oocyst and liver stages

We generated a parasite line expressing *Pb*HscB tagged with GFP at its C-terminus under the control of its endogenous promoter. To accomplish this, we transfected the *Pb*HscB-GFP tagging vector (S1A Fig) into blood stage schizonts using a standard transfection method [47]. After positive selection with pyrimethamine, we confirmed correct integration of the construct by PCR analysis (S1B Fig). Furthermore, the expression of the *Pb*HscB-GFP fusion protein in asexual blood stage parasites was assessed using flow cytometry (S1C Fig). To investigate the localization of *Pb*HscB-GFP during the oocyst stage, a mouse was infected with *Pb*HscB-GFP-expressing parasites and female *Anopheles stephensi* mosquitoes were allowed to feed on the anesthetized mouse. On day 12 post-feeding, we extracted midguts from the mosquitoes and stained them with the mitochondrial dye MitoView 650, while DNA was stained with Hoechst

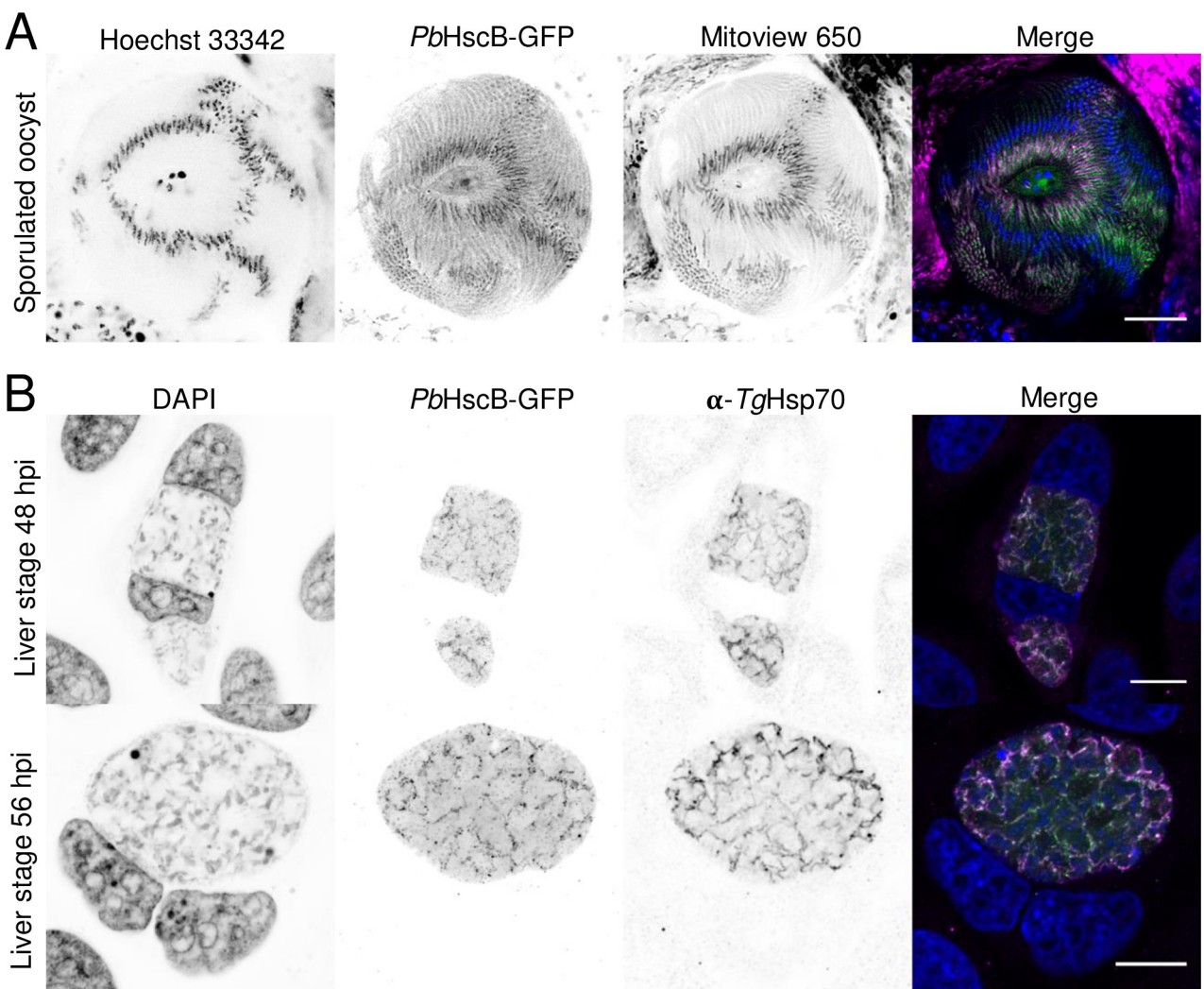

**Fig 1. *Pb*HscB-GFP localizes to the mitochondrion in the oocyst and liver stages. (A)** Mosquito midguts infected with *Pb*HscB-GFP expressing parasites were extracted on day 12 post-feed, stained with Hoechst 33342 (DNA) and Mitoview 650 (mitochondrion), and imaged live by confocal microscopy. Nuclei are shown in blue, *Pb*HscB-GFP in green, and mitochondrion in magenta. The scale bar is 10 μm. **(B)** HeLa cells were infected with *Pb*HscB-GFP expressing sporozoites and fixed at 48 and 56 hpi. Cells were stained with DAPI (DNA, blue), antibodies against GFP (green), and against *Tg*Hsp70 (magenta) and imaged by confocal microscopy. Scale bars are 10 μm.

33342. Live confocal microscopy imaging revealed overlapping signals between *Pb*HscB-GFP and the mitochondrial dye, indicating that *Pb*HscB-GFP localized to the mitochondria at this stage (Fig 1A). To study the localization of *Pb*HscB-GFP during liver stage development, we extracted *Pb*HscB-GFP-expressing sporozoites from the salivary glands of infected mosquitoes and used them to infect HeLa cells. The cells were fixed at 48 and 56 hours post-infection (hpi) and stained with antibodies against GFP and *Toxoplasma gondii* Hsp70 (*Tg*Hsp70), which has been shown to recognize *P. berghei* mitochondrial Hsp70 [48]. Although *Pb*HscB-GFP expression was weak, a clear overlap between the *Pb*HscB-GFP signal and the mitochondrial marker *Tg*Hsp70 was observed at both time points (Fig 1B and S1D Fig). This finding confirmed the expected localization of *Pb*HscB to the mitochondria during these stages of the parasite's life cycle.

## *Pb*HscB is dispensable for oocyst formation and development

To investigate the role of *Pb*HscB in parasite development, a clonal *Pb*HscB-KO line was generated. This was achieved by transfecting blood stage schizonts with a *Plasmo*GEM knockout vector (*Pb*GEM-239805) [49] targeting the PBANKA_0821000 locus using a standard transfection method [47]. The *Plasmo*GEM vector was designed to undergo double crossover homologous recombination, resulting in its integration at the correct locus (S1E Fig). To obtain a clonal knockout parasite line, transgenic parasites were subjected to limiting dilution. This process yielded three clones, and PCR analysis (S1F Fig) confirmed the presence of the knockout construct and the absence of the endogenous locus in all three clones. Clone 1 (Cl.1) was selected for further analysis. Since viable oocysts and sporozoites are a prerequisite for the investigation of parasite liver stage development, we investigated oocyst development and sporozoite generation (S2 Fig). For this purpose, we allowed female *A. stephensi* mosquitoes to feed on anaesthetized mice that had a suitable number of gametocytes in their blood. Mosquito midguts infected with either WT or *Pb*HscB-KO parasites, were dissected and imaged on day 7, 10, and 14 post-feeding to study oocyst development. *Pb*HscB-KO oocysts exhibited a similar size as WT at day 7, 10 and 14 post-feeding (S2A Fig). Detailed imaging at higher magnification revealed normal sporozoite formation on day 14 post-feeding (S2B Fig, white arrows). Importantly, *Pb*HscB-KO parasites produced normal sporozoite numbers in the salivary glands of the mosquitoes (see below). These findings collectively suggest that *Pb*HscB is not essential for oocyst development and sporozoite formation.

## *Pb*HscB-KO strongly impairs liver stage development *in vitro* and *in vivo*

To assess exo-erythrocytic or liver stage development, *Pb*HscB-KO and WT sporozoites were used to infect HeLa cells *in vitro*. The size of parasites was measured using automated microscopy from 6 to 56 hours post-infection (hpi) and revealed a significant growth defect of *Pb*HscB-KO parasites (Fig 2A). *Pb*HscB-KO parasites appeared smaller compared to WT at 24, 48 and 56 hpi. However, it is worth noting that the KO parasites were still able to grow over time. Comparison of the parasite numbers at 6 and 48 hpi indicated reduced survival in the *Pb*HscB-KO parasites (Fig 2B). Previous studies have linked reduced survival of WT parasites during liver stage development to host cell autophagy [50, 51]. To investigate whether *Pb*HscB-KO parasites might be impaired in shedding the autophagy marker microtubule-associated protein 1 light chain 3 (LC3) from their parasitophorous vacuole membrane (PVM), WT and *Pb*HscB-KO parasites were fixed at 24 and 48 hpi and stained with LC3 antibodies. At 24 hpi, both WT and *Pb*HscB-KO parasites were heavily decorated with LC3. However, at 48 hpi, approximately half of the WT parasites were able to shed the autophagy marker, while the majority of KO parasites remained LC3-positive (Fig 2C). This reduced shedding capacity of the *Pb*HscB-KO parasite correlates with the reduced survival observed in Fig 2B and could explain the lower number of *Pb*HscB-KO parasites present at 48 hpi. Due to the small size of *Pb*HscB-KO parasites we questioned whether they are still able to initiate late liver stage development. Therefore, infected cells were fixed at 56 hpi for WT parasites and at 96 hpi for *Pb*HscB-KO parasites. Cells were stained with antibodies against merozoite surface protein 1 (MSP1), which is known to be only expressed at late liver stage (from 48 hpi onwards). MSP1 was readily detected at 56 hpi in WT parasites (Fig 2D), while it was weakly detected at 72 hpi (not shown) and more strongly detected at 96 hpi in *Pb*HscB-KO parasites (Fig 2D). This observation indicates a significant delay in liver stage development for *Pb*HscB-KO parasites but also suggests that, although delayed, these parasites might be capable of completing their liver stage development. To assess if parasites lacking *Pb*HscB can indeed complete their liver stage and establish a blood stage infection, C57BL/6 mice were infected with either 5,000 WT

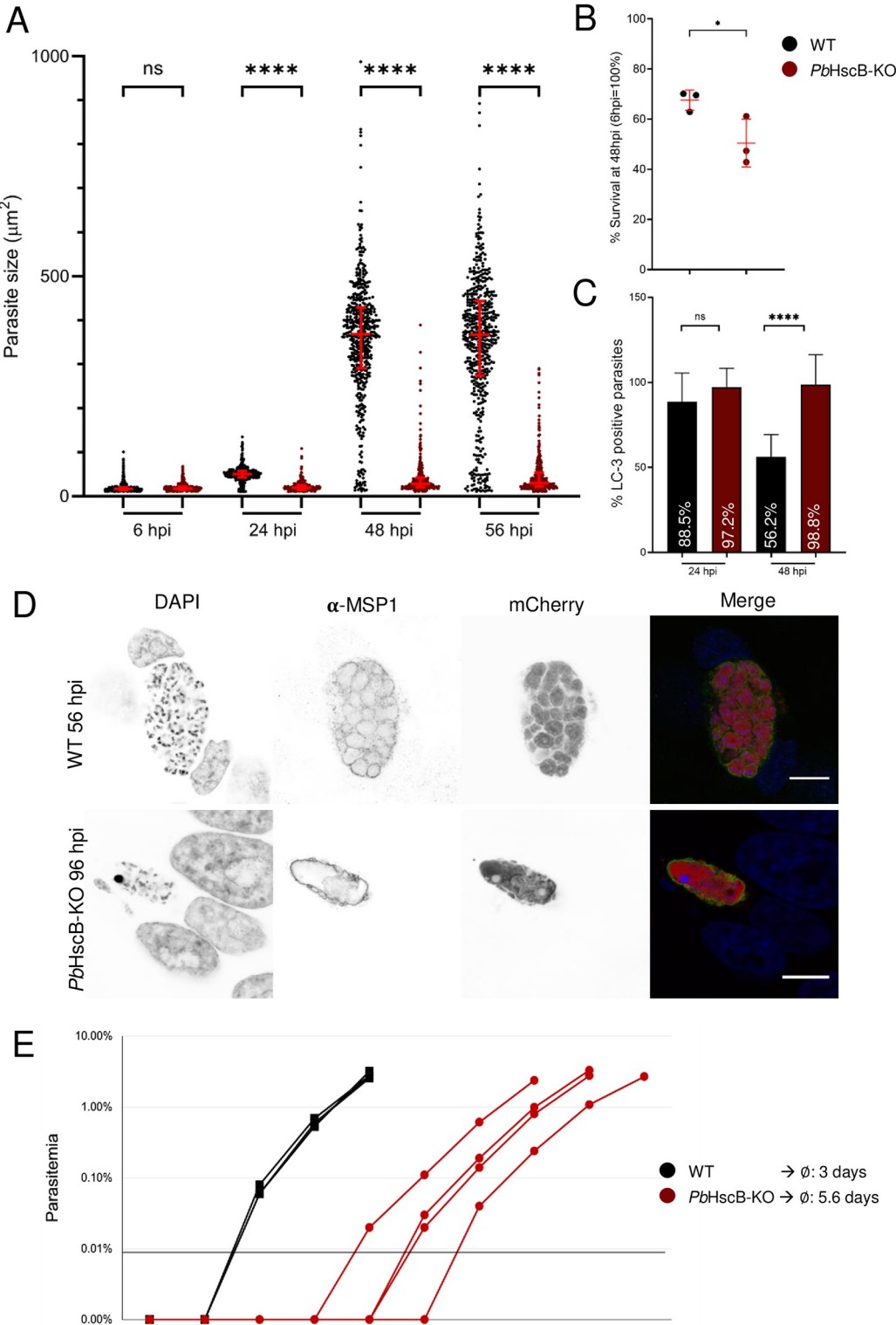

**Fig 2. *Pb*HscB-KO strongly impairs liver stage development *in vitro* and *in vivo*. (A)** HeLa cells were infected with either WT or *Pb*HscB-KO sporozoites and imaged at indicated time points using automated microscopy. Parasite size of 500 parasites each was measured based on cytosolic mCherry signal expressed by the parasites. The significance of the results was determined with Kruskal-Wallis test; p-values that are below 0.0001 are presented with ****. The result shown represents one out of three experiments. **(B)** Parasite number was determined at 6 hpi and 48 hpi by automated

microscopy. The number of parasites at 6 hpi was set to 100%. Significance was calculated by unpaired t-test; p = 0.0461 and is presented as * (p < 0.05). Each dot represents an infection experiment. **(C)** WT- or *Pb*HscB-KO-infected HeLa cells were fixed at 24 and 48 hpi and stained with anti-LC3B antibodies. LC3-staining on PVM was quantified manually at 24 and 48 hpi (n = WT 24 hpi: 200; KO 24 hpi: 107; WT 48 hpi: 137; KO 48 hpi: 81). Significance of results was determined using unpaired t-test; **** represents p < 0.0001. **(D)** Infected cells were fixed at 56 hpi for WT and 96 hpi for *Pb*HscB-KO parasites and stained with DAPI (DNA, blue) and anti-MSP1 antibodies (parasite plasma membrane, green). Cytoplasmic mCherry is shown in red. Scale bars are 10 μm. **(E)** 5,000 sporozoites per mouse were injected intravenously into C57BL/6 mice per parasite line in two independent experiments. Represented is one experiment (4 mice for WT and 4 mice for *Pb*HscB-KO parasites). Parasitemia of mice was monitored daily by flow cytometry. A parasitemia of 0.01% (100 parasites per 1 million erythrocytes) was set as a threshold. The average prepatent period was calculated for both parasite lines (WT: 3 days; *Pb*HscB-KO: 5.6 days).

or *Pb*HscB-KO sporozoites. Parasitemia was monitored daily by flow cytometry, and a threshold of 100 parasites per 1 million erythrocytes (≙ 0.01%) was set to classify a mouse as blood stage positive. All mice infected with WT parasites were blood stage parasites positive on day 3 post-injection, while mice infected with KO parasites were positive on days 5 to 7 post-injection. On average, *Pb*HscB-KO parasites exhibited a delay in prepatency of 2.6 days compared to WT (Fig 2E). This represents a significant impairment of liver stage development; however, it also demonstrates that *Pb*HscB-KO parasites are still capable of establishing a blood stage infection.

This result demonstrates that *Pb*HscB-KO is not sufficient for generating a single knockout GAP but the significant delay in liver stage development exhibited by *Pb*HscB-KO parasites makes *Pb*HscB an interesting candidate for a double gene deletion approach to generate a safe attenuated parasite vaccine strain.

## *Pb*HscB-*Pb*Mei2-dKO parasites show normal oocyst development and sporozoite production

As a second gene deletion, PlasMei2 (*Pb*Mei2) was chosen, as this gene was shown to be essential for liver stage completion in *P. yoelii* [31]. To generate a marker-free *Pb*HscB-KO line, negative selection was used (S3A and S3B Fig). Subsequently, the *Plasmo*GEM vector for *Pb*Mei2-KO (*Pb*GEM-300555) [49] was transfected into the marker-free *Pb*HscB-KO schizonts as performed in [47] (S3C Fig). Correct integration of the construct was confirmed by PCR analysis (S3D Fig). The transgenic parasites were subjected to limiting dilution to obtain clonal parasite lines resulting in four clones with the correct knockout of the *Pb*Mei2 locus (S3D Fig). Clone 1 (Cl.1) was selected for further investigation. Initially, oocyst development was analyzed as described previously. In addition to the *Pb*HscB-*Pb*Mei2-double knockout (dKO) parasite line, a *Pb*Mei2 single knockout (sKO) line was also included for comparison. No significant difference in oocyst size was observed between WT, *Pb*HscB-KO, *Pb*Mei2-KO and *Pb*HscB-*Pb*Mei2-dKO lines at day 7 and day 14 post-feeding, indicating that both gene deletions are dispensable for oocyst development (Fig 3A). This finding was further supported when counting the number of sporozoites present in salivary glands at earlier (day 18 and 19 post-feeding) and later (day 24, 26, and 28 post-feeding) time points, which showed no significant difference among the WT and the three KO lines (Fig 3B). Thus, all three KO lines, including the *Pb*HscB-*Pb*Mei2-dKO line, produce normal numbers of salivary gland sporozoites, an important prerequisite for the generation of attenuated parasites as vaccine strains.

## *Pb*HscB-*Pb*Mei2-dKO parasites are fully arrested during late liver stage development

To investigate liver stage development of *Pb*HscB-*Pb*Mei2-dKO parasites, HeLa cells were infected with sporozoites from WT, *Pb*HscB-KO, *Pb*Mei2-KO and *Pb*HscB-*Pb*Mei2-dKO

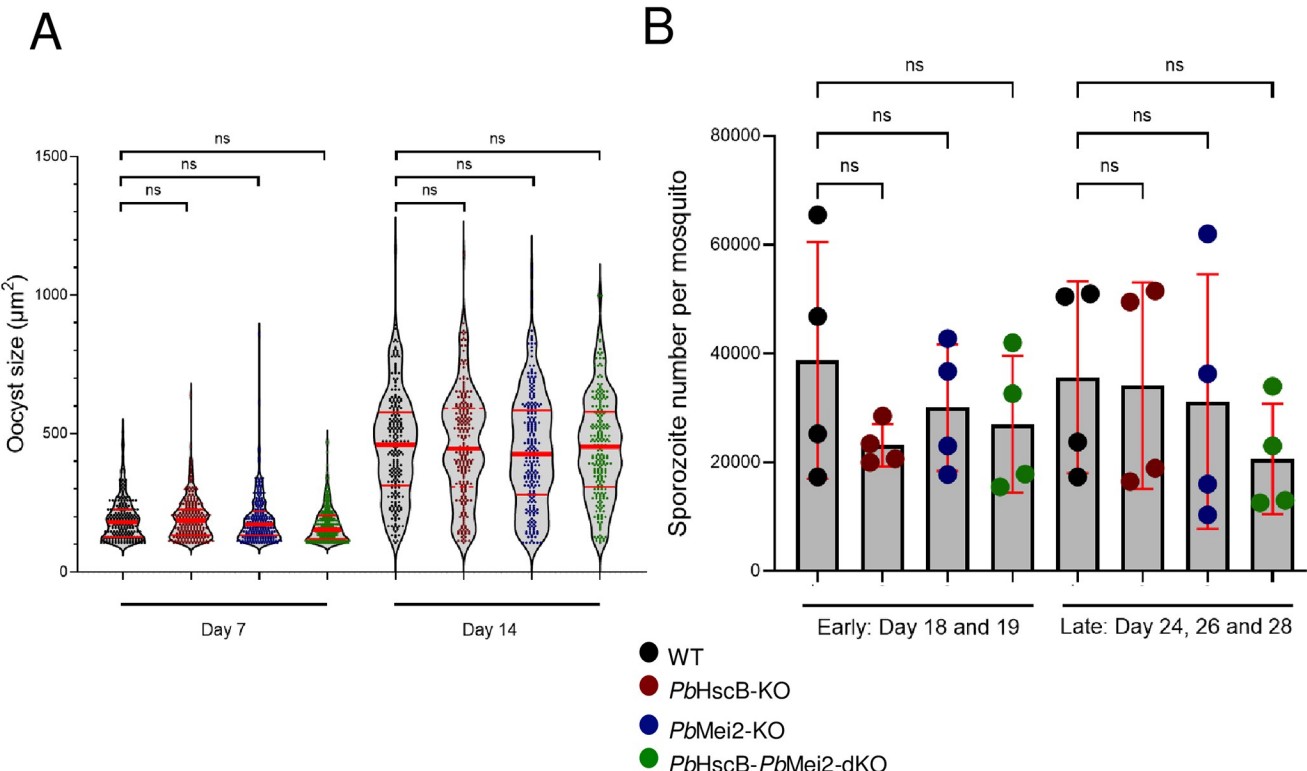

**Fig 3. *Pb*Mei2-KO and *Pb*HscB-*Pb*Mei2-dKO do not affect oocyst formation and sporozoite production. (A)** Midguts of infected mosquitoes were dissected (n = 10) and imaged on day 7 and day 14 post-feeding. Images were acquired using a 5x objective and analyzed using ImageJ software: mCherry signal was binarized using the threshold function and oocyst size was measured with the 'analyze particles function'. The presented result is a representative of three experiments and shows 240 oocysts per parasite line as violin plot with individual oocyst sizes and medians with interquartile ranges. Kruskal-Wallis test did not result in a significant difference (ns) between WT and modified parasite line oocysts in neither at day 7 nor at day 14 post feeding. ($p_{day7\ 1868-dKO}$ = 0.5731, all others >0.9999). **(B)** Infected salivary glands were dissected on indicated time points, homogenized and the number of sporozoites was determined with a Neubauer counting chamber. Each data point represents the mean of salivary gland sporozoites number from 10 mosquitoes determined in four independent experiments. The significance of the results was calculated using the One-way Anova test.

parasites. As previously shown, automated microscopy analysis revealed that *Pb*HscB-KO parasites exhibited impaired growth at 24, 48 and 56 hpi, while *Pb*HscB-*Pb*Mei2-dKO parasites showed an even more pronounced growth defect (Fig 4A). *Pb*HscB-KO parasites consistently showed a significant reduction in size at all time points. When both knockouts were combined in the *Pb*HscB-*Pb*Mei2-dKO parasites, the phenotype resembled that of *Pb*HscB-KO, displaying a substantial reduction in size and slow growth towards 48- and 56-hours post-infection (hpi) (Fig 4A). To assess parasite survival, the parasites were counted at 48 hpi and compared to the count at 6 hpi. This analysis revealed that the three mutant lines showed reduced survival compared to the WT parasites, with the *Pb*HscB-KO parasites exhibiting a significant reduction in survival (Fig 4B). Since *Pb*Mei2-KO parasites appeared to grow to a normal size, as previously reported for *P. yoelii* Mei2-KOs [31], further investigation was conducted on the late liver stage. *Pb*Mei2-KO-infected cells were fixed at 56 hpi and stained for MSP1, which is expressed just prior to merozoite formation [52], and for mitochondria using anti-*Tg*Hsp70 antibodies to assess parasite fitness. While WT parasites displayed cytomere stages with the invaginated plasma membrane and a strong MSP1 signal, *Pb*Mei2-KO parasites showed neither plasma membrane invaginations nor distinct MSP1 signal (Fig 4C). This finding confirms the observations made by Dankwa and colleagues for *P. yoelii* Mei2-KO parasites [31]. Our findings

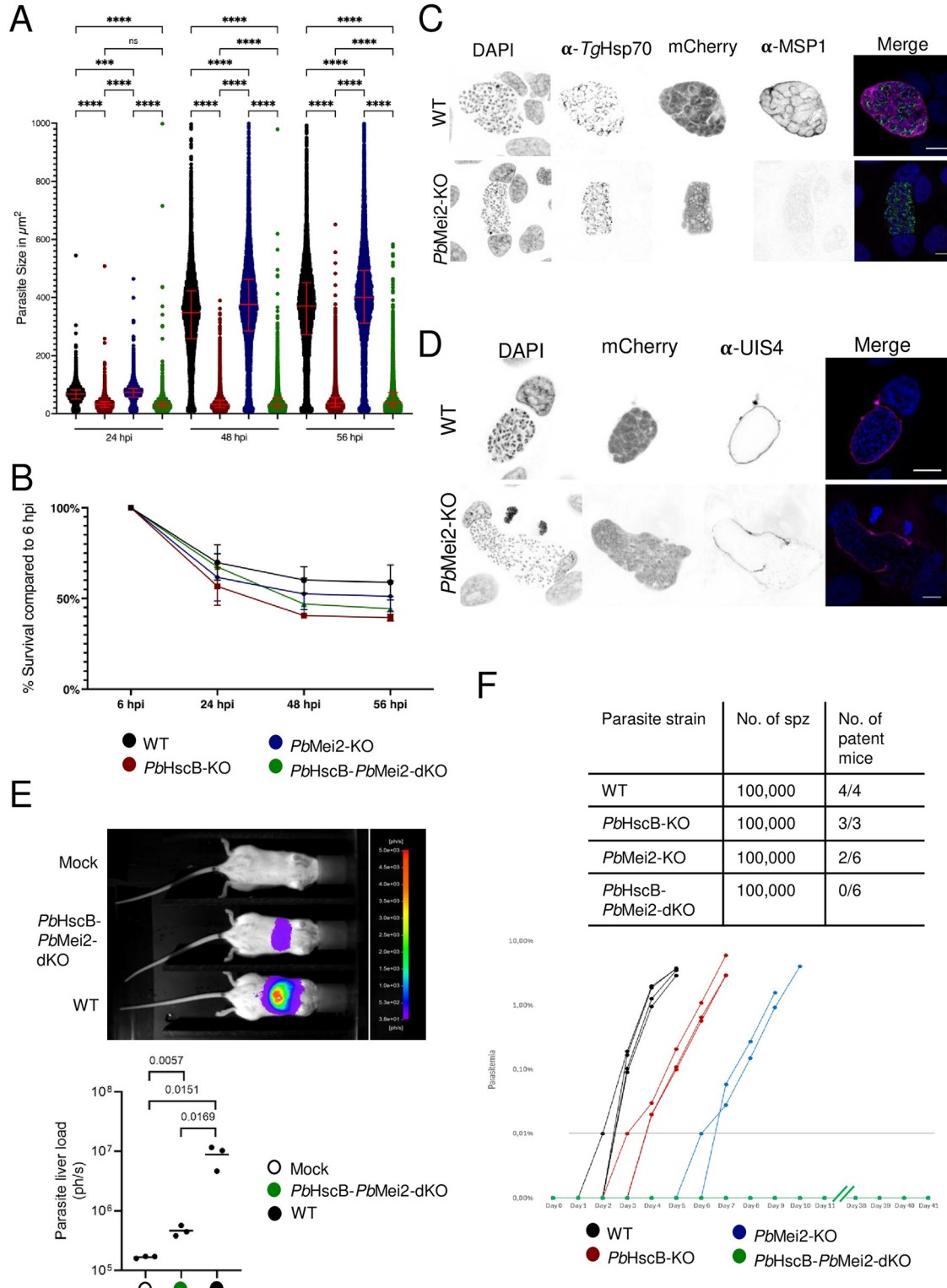

**Fig 4. *Pb*Mei2-KO and *Pb*HscB-*Pb*Mei2-dKO parasites are arrested in the liver stage. (A)** HeLa cells were infected with either WT or *Pb*HscB-KO sporozoites and imaged at indicated time points using automated microscopy. The shown result is a representative of three experiments and shows parasite size of 500 parasites measured based on cytosolic mCherry signal expressed. The significance of the results was determined with Kruskal-Wallis test; (****: p<0.0001, ns: $p_{24h1868-Mei2KO}$ = 0.9528, $p_{48h1868-Mei2KO}$ = 0.076, $p_{56h1868-Mei2KO}$ = 0.0674). **(B)** The number of parasites was determined at 6 hpi and 48 hpi by automated microscopy and the number

at 6 hpi was considered as 100%. Significance was calculated by Kruskal-Wallis test, with Dunn's multiple comparison test (*: p = 0.0276; ns: $p_{1868-Mei2KO} > 0.9999$, $p_{1868-dKO} = 0.4231$). Each dot represents an infection experiment. **(C)** HeLa cells infected with WT or *Pb*Mei2-KO parasites were fixed at 56 hpi and stained with DAPI (DNA, blue), anti-*Tg*Hsp70 antibodies (mitochondrion, green), and anti-MSP1 antibodies (parasite plasma membrane, magenta). Cytosolic mCherry signal is shown in red. Scale bars are 10 μm. **(D)** HeLa cells infected with WT or *Pb*Mei2-KO parasites were fixed at 56 hpi and stained with DAPI (DNA, blue) and antibodies against UIS4 (PVM, magenta). Cytosolic mCherry is shown in red. Scale bars are 10 μm. **(E)** Albino mice (B6(C)/Rj-Tyrc/ c) injected with 100,000 sporozoites from mosquito salivary glands infected with either *Pb*HscB-*Pb*Mei2-dKO or WT parasites were assessed for luciferase activity via luminescence measurement. Salivary glands from uninfected *P. berghei* mosquitoes served as a negative control (Mock). At 44 hpi mice were anaesthetized, injected with luciferin and parasite load in the liver was analyzed using an In Vivo Imaging System (IVIS). The graph displays the mean total photon counts per second, with each dot representing an individual mouse (N = 3 per group). Statistical significance was determined using an unpaired t-test. **(F)** C56BL/6 mice were infected with 100,000 sporozoites each (4 mice for WT, 3 mice for *Pb*HscB-KO, 4 mice for *Pb*Mei2-KO and 6 mice for *Pb*HscB-*Pb*Mei2-dKO). Parasitemia was monitored daily by flow cytometry and a parasitemia of 0.01% (100 parasites per 1 million erythrocytes) was set as threshold. In a pilot study, it was already confirmed that infection of mice with 100,000 *Pb*Mei2-KO sporozoites resulted in breakthrough infections.

indicate that the single knockout *Pb*Mei2-dKO and the double knockout *Pb*HscB-*Pb*Mei2-dKO parasites do not express MSP1, suggesting they do not reach the very late stages of liver development where this protein is typically produced. Additionally, the mitochondria exhibited abnormal accumulations, and the cytoplasmic mCherry signal revealed numerous vacuoles within the parasite (Fig 4C). These findings indicate that the parasite is no longer developing normally. Some *Pb*Mei2-KO parasites also displayed an aberrant shape, which could result from issues related to the maintenance of the PVM. Indeed, staining with the PVM marker UIS4 showed that *Pb*Mei2-KO parasites at 56 hpi had a corrupted PVM and an apparent leakage into the host cell cytosol was observed (Fig 4D). This observation could also explain the slight increase in parasite size measured at 48 and 56 hpi (Fig 4A), as parasites lacking an intact PVM might expand to occupy more space within the host cell cytosol. Further investigation is required to understand why the PVM breaks down and the role of *Pb*Mei2 in this process.

As *Pb*Mei2-KO [32] and *Pb*HscB-KO (this study) have shown breakthrough infections in mice, we verified whether the double-KO line would be completely arrested during liver stage development. First, we confirmed that the *Pb*HscB-*Pb*Mei2-dKO line could invade and develop in mouse hepatocytes. Using luminescence imaging, we monitored *in vivo* liver stage manifestation after injecting 100,000 WT or *Pb*HscB-*Pb*Mei2-dKO sporozoites, while salivary glands from uninfected mosquitoes served as a negative control (Mock). At 44 hpi, although the luminescence signal was reduced relative to WT, most likely due to the smaller parasite size (Figs 2A and 4A), it was still detectable in mouse livers. Following confirmation of liver presence, we assessed potential breakthrough infections by injecting 100,000 sporozoites from WT, *Pb*HscB-KO, *Pb*Mei2-KO, or *Pb*HscB-*Pb*Mei2-dKO lines into mice and tracking parasitemia over time. As anticipated, blood-stage parasitemia appeared with a 2–3 day delay in *Pb*HscB-KO and a 5–6 day delay in *Pb*Mei2-KO sporozoite-injected mice. Importantly, no breakthrough infections were observed in mice injected with *Pb*HscB-*Pb*Mei2-dKO sporozoites for up to 41 days post-injection (Fig 4F). This demonstrates that the *Pb*HscB-*Pb*Mei2-dKO line is robustly blocked during *P. berghei* liver stage development under these conditions. The next step would be to test this approach in *P. falciparum*, as both genes, HscB and Plasmei2, are present in the genome of the human parasite. This strategy of fully attenuating *P. falciparum* by simultaneously knocking out *Pf*HscB and *Pf*Mei2 could potentially result in the development of a safe and effective malaria vaccine.

## Discussion

This study underscores the significance of targeting distinct pathways in *Plasmodium* liver stage development to create safer GAPs. Our work demonstrates that the deletion of *Pb*HscB

in combination with *PlasMei2* (*Pb*HscB-*Pb*Mei2-dKO) effectively arrests liver stage progression, preventing breakthrough infections under experimental conditions. This double-knockout strategy builds on prior findings that single gene deletions, while partially effective, may not fully prevent parasite escape and blood stage development [32].

The double deletion approach addresses critical concerns raised by single-knockout models such as *PbPlasMei2*-KO and *PbHscB*-KO, which have shown delayed blood stage emergence but potential for breakthrough under certain conditions [32]. The robustness of the *Pb*HscB-*Pb*Mei2-dKO line in halting liver stage progression highlights the importance of disrupting multiple, non-redundant pathways to achieve complete developmental arrest. By combining deletions affecting mitochondrial iron-sulfur cluster biogenesis and RNA-binding processes critical for the parasite's final replication stage, we effectively block a pivotal point in liver stage development.

It has been demonstrated that the parasite relies on the apicoplast SUF pathway for ISC biogenesis during blood stage development and sporogony [45, 46]. Our findings indicate that the parasite relies on the SUF pathway for ISC generation during both the blood and mosquito stages, while it requires the mitochondrial ISC pathway only during liver stage development. The loss of *Pb*HscB and potential loss of sufficient ISC biogenesis have a significant impact on liver stage growth. However, since some *Pb*HscB-depleted parasites are still able to complete liver stage development, the protein is apparently important but not essential for this parasite stage. The observed reduced growth phenotype of *Pb*HscB-KO parasites may result from impairment in different metabolic pathways due to limited ISC availability.

One example of a metabolic pathway that could be affected is the electron transport chain (ETC) in the mitochondrial inner membrane, which contains several Fe-S proteins, including succinate dehydrogenase [53]. The pyrimidine synthesis pathway is highly dependent of the ETC via the dihydroorotate dehydrogenase [54]. Consequently, issues in the ETC due to lacking ISCs could explain the reduced growth rate in the liver.

Another crucial pathway for liver stage development is the *de novo* synthesis of heme. During intra-erythrocytic development, the parasite can scavenge heme from the host cell. However, during exo-erythrocytic development, the parasite relies on *de novo* heme synthesis [55]. The impairment of ISC biogenesis in the absence of *Pb*HscB could potentially affect the availability of ISCs for heme synthesis, leading to further disruptions in liver stage growth. The presence of an iron-response element (IRE) in the mRNA of δ-aminolevulinic acid synthetase (ALAS), the first enzyme in the *de novo* heme biosynthesis pathway, suggests a potential link between ISC availability and heme biosynthesis regulation during mosquito and liver stage development [56]. Aconitase, another enzyme involved in ISC metabolism, typically contains an ISC to carry out its catalytic function. However, under conditions of limited ISC availability, aconitase lacking bound ISC can repress the translation of IRE-containing transcripts [57, 58]. In the context of *Pb*HscB-KO parasites, if ISC concentration is low, it is possible that the translation of ALAS, which contains an IRE, could be repressed. This repression could lead to impaired heme biosynthesis, thereby affecting liver stage development. Further investigations and experimental studies are needed to validate this hypothesis and explore the precise molecular mechanisms underlying the role of *Pb*HscB in ISC assembly and its potential influence on heme biosynthesis regulation during liver stage development. By unraveling these connections, we can enhance our understanding of the intricate processes involved in *Plasmodium* parasite biology and potentially identify new targets for intervention strategies against malaria.

The studies on PlasMei2 have provided compelling evidence for its crucial role in liver stage development of *Plasmodium* parasites, including *P. yoelii*, *P. berghei* and *P. falciparum* [31–33]. Deletion of PlasMei2 in *P. yoelii* and *P. falciparum* resulted in an arrest of the parasite just before merozoite formation [31–33] indicating its crucial function for late liver stage

development. In the case of *Pb*Mei2-KO parasites, it was observed that PVM integrity is lost at approximately 56 hpi, leading to leakage into the host cell cytosol (Fig 4D).

The exact function of PlasMei2 and its underlying molecular mechanisms is unknown. The presence of an RNA-binding motif in PlasMei2 suggests its involvement in translational regulation [31]. Translational regulation has been demonstrated to be important for stage transitions in the parasite's life cycle, particularly during the sporozoite to trophozoite transition [26, 27, 59–61]. We speculate that PlasMei2 may play a role in regulating specific transcripts that are critical for initiating liver stage merogony, the process of daughter cell formation. Depletion of PlasMei2 results in the failure to initiate daughter cell formation in most mutants, as indicated by the absence of membrane invaginations during cytomere stage and the failure to express MSP1 on the parasite plasma membrane. Because of being unable to progress in their development, the PVM integrity might ultimately be lost.

To further elucidate the function of PlasMei2 and improve the safety profile of Plas-Mei2-KO parasites as GAPs, it is essential to identify its potential mRNA targets. This identification could offer valuable insights into the mechanisms governing the parasite's transition from proliferation to daughter cell formation, a process that remains poorly understood. Confirming PlasMei2's role in translational regulation would deepen our understanding of the complex molecular processes involved in *Plasmodium* development and could potentially reveal new strategies for malaria control.

The safety and efficacy of PlasMei2-KO parasites as GAPs have been assessed in various experimental models, including *P. yoelii* in BALB/c mice and *P. falciparum* in humanized mice [31, 33]. However, given that breakthrough infections have been observed with rodent *mei2*-KO parasites in our study and others [32], a single *mei2* knockout strategy may not be sufficiently safe for the human parasite *P. falciparum* as well, especially in immunocompromised individuals. This underscores the need for additional genetic attenuations to enhance safety. In addition, previous research [35], has demonstrated exceptionally low breakthrough rates in GAPs. This highlights the importance of considering such results when evaluating breakthrough infections, especially when using a limited number of animals for testing.

Combining PlasMei2-KO with a second gene deletion that has a different function might increase the likelihood of targeting distinct crucial developmental pathways within the parasite, thus avoiding redundancy in protein function and preventing the parasite from overcoming the impaired pathway. In a recent study, simultaneous deletion of the *PlasMei2* and *linup* genes was employed to achieve parasite attenuation [36]. The resulting double-attenuated parasite lines, *Py*LARC2 and *Pf*LARC2, demonstrated either no expression (*Py*LARC2) or very weak expression (*Pf*LARC2) of MSP1. This limited MSP1 expression raises questions about the generation of protective immune responses against late liver and asexual blood-stage parasites. Potentially breakthrough infections then lead to a full blood-stage infection. This underscores the importance of considering the stage-specific expression of protective antigens when evaluating the efficacy and safety of genetically attenuated parasites. Moreover, as the authors pointed out, both proteins, PlasMei2 and LINUP, might act in concert to regulate gene expression and may thus have partly redundant function. It is therefore crucial to identify alternative pathways or functions for generating safely attenuated parasites. In our study, *Pb*HscB-*Pb*Mei2-dKO parasites perfectly fulfill this latter criterion. *In vitro* studies revealed that the *Pb*HscB-*Pb*Mei2-dKO parasites exhibit a similar phenotype to the *Pb*HscB-KO parasites, displaying a reduced growth rate. This reduced growth rate may be beneficial for generating a strong and long-lasting protective immune response. PlasMei2-KO parasites grow normally until the end of schizogony, potentially allowing the host cells to present a wider array of parasite epitopes via the major histocompatibility complex I (MHC-I) pathway compared to early arresting GAPs. However, given that our newly generated *Pb*HscB-*Pb*Mei2-dKO parasites also

do not express MSP1, we do not anticipate an advantage in this regard. It will now be important to determine whether *P. falciparum* HscB-Mei2-dKO parasites express MSP1 to fully assess their potential for inducing protective immunity.

The observed breakdown of the PVM in *Pb*Mei2-KO parasites indicates that the parasites may be eliminated faster following schizogony. The slow growth phenotype of *Pb*HscB-*Pb*Mei2-dKO parasites may enable them to persist in the liver for a longer period, potentially providing better long-term protection [62]. *Pb*HscB-*Pb*Mei2-dKO parasites may prolong the period during which hepatocytes can present parasite antigens, which has been demonstrated to be important for priming parasite-specific CD8$^+$ T cells and generating a robust immune response [63, 64]. For this work, we decided against performing challenge experiments, as the goal of this study was to demonstrate that deleting completely unrelated parasite pathways still allows parasites to develop until the late liver stage but prevents breakthrough infections. The next objective is to test this in *P. falciparum* to generate a potent and safe LARC GAP and evaluate its protective efficacy in preclinical and clinical studies.

In conclusion, this study highlights that targeting divergent yet essential pathways in the *Plasmodium* liver stage can create a strong block against parasite progression. The *Pb*HscB-*Pb*Mei2-dKO parasite line sets the stage for further exploration into multigene knockouts as a strategic pathway for safe, effective malaria vaccine development. Ongoing and future preclinical studies will be pivotal to validate this approach in human models, advancing the fight against malaria.

## Experimental procedures

### Animal experiments conducted at the University of Bern

Animal experiments were carried out in strict accordance with the guidelines of the Swiss Tierschutzgesetz (TSchG; Animal Rights Laws) and approved by the ethical committee of the University of Bern (Permit Number: BE98/19). Mice were kept in specific-pathogen-free conditions and screened regularly for pathogens. They were housed in individually ventilated cages with autoclaved aspen woodchip, a mouse house and paper tissue at a temperature of $21 \pm 2°C$ with a 12:12 h light-dark cycle and a relative humidity of $55 \pm 10\%$. Mice were fed *ad libitum* with commercially prepared and autoclaved dry rodent pellets and water. Health of mice was monitored by routine daily visual health checks. The parasitemia of infected animals was measured by flow cytometry or by blood smears.

BALB/c mice (6–8 weeks; Janvier Laboratories, France) were used to maintain parasites and for mosquito feeds. Mice were infected with parasites by intraperitoneal or intravenous injection. When parasitemia reached 2–5%, mice were euthanized by $CO_2$ and parasitized blood was isolated by heart puncture. For feeding of mosquitoes, mice were anaesthetized upon reaching a parasitemia of 5–7% with sufficient gametocyte number. Anesthesia was carried out with ketamine:xylazine and when mice did no longer react to touch stimulus, were placed on cages with approximately 150 mosquitoes.

Female C57BL/6 mice (6–8 weeks; Janvier Laboratories, France) were used to conduct *in vivo* studies of liver development (see "*in vivo* experiments" in this section for mice and salivary gland sporozoites number used and duration of experiment). For this, mice were infected intravenously with isolated salivary gland sporozoites. When parasitemia reached 2%, this was considered as the end of the experiment and mice were euthanized by $CO_2$ and parasitized blood was isolated by heart puncture to perform further genotypical analysis.

We were careful in verifying *Plasmodium* infected animals during the parasite maintenance and *in vivo* experiments with signs where animals have to be euthanized as: abnormal

behaviors, pathological changes, ruffled fur, reduced mobility. During these experiments no animals infected with parasites were found dead before meeting criteria for euthanasia.

## Parasite strains

All parasite strains used were of *P. berghei* ANKA background that expressed cytosolic mCherry under the hsp70 promoter and luciferase under the eEF1α promoter (*Pb*1868). These *Pb*1868 parasites are phenotypically similar as *Pb*WT [50] and were used as background line for the GFP and knockout lines that were generated.

## Generation and selection of transgenic parasites

To generate an endogenously GFP-tagged parasite line for *Pb*HscB, the C-terminal part of the gene PBANKA_0821000 was amplified (Fwd_Primer: AAggtacccaatccaattcatagATAGTAATAAATC/ Rev_Primer: AAgggcccTATATTTTGCAACCTATCC) and cloned into a vector containing a GFP and hdhfr as selection marker (resulting in a C-term HscB-GFP fusion protein). The construct was linearized with *Eco*RV restriction digest. To generate a *Pb*HscB-KO parasite line, a knockout construct was provided by PlasmoGEM (PbGEM-239805) [49]. The constructs were transfected into blood stage *Pb*1868 schizonts by a standard procedure [47]. In short, a mouse was infected with *Pb*1868 parasites and blood was extracted once parasitemia reached 1–3%. The infected blood was cultured in schizont culture medium (Roswell Park Memorial Institute RPMI1640 medium supplemented with 20% fetal calf serum (FCS)) under 5% $CO_2$, 5% $O_2$ and 90% $N_2$. Parasites were cultured over night at 36.5˚C with constant rotation. The following day, schizonts were isolated by density gradient centrifugation. For this, the schizont culture was underlaid with a 60% Nycodenz in PBS mixture. Parasites were centrifuged for 20 min at 450 g with minimal acceleration and deceleration. The resulting brownish schizont layer was harvested and centrifuged in RPMI1640 for 8 min at 450 g. The schizont pellet was resuspended with 100 µL transfection buffer (Amaxa® Human T cell Nucleofector® Kit) supplemented with 10 µg of either *Eco*RV-digested *Pb*HscB-GFP vector or *Not*I-digested *Pb*GEM-239805. The schizont-DNA mixture was transferred to a transfection cuvette and electroporated with an Amaxa Nucleofector 2b device using program U33. After immediate addition of 100 µL of fresh RPMI1640 the electroporated parasites were injected intravenously into BALB/c mice. The next day, pyrimethamine (70 µg/L) was added to the drinking water. When parasitemia of infected mice reached 3%, mice were euthanized by $CO_2$ and blood was extracted by heart puncture. Genomic DNA was extracted, and presence of the constructs was determined by PCR. Primers used for integration of GFP-construct were Primer 1: ATGAATAAAATTAAATATGCATTTCC and Primer 2: TCGATGCCCTTCAGCTCG (S1A Fig). Primers used for *Pb*HscB-KO construct were Primer 3: catcatttaactgttgtaatatattgtttatactc; Primer 2: gttacttagttacccaacaatccg and Primer 5: cataatctggaacatcatatggatac (S1E Fig). KO parasites were subjected to limiting dilution. For this, a mouse was infected with transgenic parasites. Once parasitemia was between 0.2 and 1%, the mouse was euthanized, and blood was collected. The number of infected erythrocytes/mL was determined by flow cytometry and parasites were diluted to inject 0.7 parasites per mouse into 10 mice. When parasitemia reached 3%, blood was again collected, and another PCR was performed to confirm presence of KO construct and absence of WT locus. Clone 1 was chosen for further analysis.

To create a second gene deletion in *Pb*HscB-KO parasites, a negative selection with 5-fluorocytosine (5-FC) was performed. For this, a mouse was infected with clonal *Pb*HscB-KO

parasites and 5-FC (1 mg/mL) was added to the drinking water after parasitemia reached 3%. Parasites were selected for one week until parasitemia recovered to around 3%. The mouse was then euthanized by $CO_2$ and blood was extracted by heart puncture. gDNA was extracted and absence of resistance cassette was verified by PCR. Primers used were Primer 3 `catcatttaactgttgtaatatattgtttatactc`; Primer 4: `gttacttagttacccaacaa tccg`; Primer 5: `cataatctggaacatcatatggatac`; Primer 6: `CAGAGAATGACCACAAC CTCTTCAG`; and Primer 7: `CAAGACTTCTGTAGCCATGATAGCAC` (S3A Fig). To obtain a clonal, marker-free KO line, the 5-FC-selected parasites were cloned by limiting dilution as describe above and absence of resistance cassette was confirmed for Clone 2 (S3B Fig). A *Pb*Mei2 (PBANKA_1122300) knockout construct provided by *Plasmo*GEM (*Pb*GEM-300555) [49] was transfected into marker-free *Pb*HscB-KO parasites as described above. Presence of knockout construct and absence of endogenous locus was checked by PCR. Primers used were Primer 8: `gcatgatgccgaatgcccaa`; Primer 9: `ccaacattacatttgttacgaaaatc aattgg` and Primer 10: `ctttggtgacagatactac` (S3C Fig). *Pb*HscB-*Pb*Mei-dKO parasites were again subjected to limiting dilution, which resulted in 4 clones (S3D Fig). Clone 1 was chosen for further analysis.

## Mosquito breeding

*Anopheles stephensi* mosquitoes were bred at 28˚C, 80% humidity and a 12:12h light-dark cycle. They were fed with cotton pads containing 8% fructose solution. Female mosquitoes were fed with human blood once a week for egg production. Eggs were kept in water baths supplemented with Liquifry fish food and hatched larvae were grown for 7–12 days until pupae were formed. Larvae were fed with fish food. Pupae were collected in water bowls and placed in mosquito cages for hatching. After feeds on parasite infected blood, mosquitoes were kept at 20˚C.

## Live imaging and quantification of midgut oocysts

Mosquitoes were aspirated with a small vacuum device and anaesthetized with chloroform. Sleeping mosquitoes were kept on petri dishes on ice. Infected midguts were extracted with forceps and placed on glass slides with 1x PBS containing 1 µg/mL Hoechst 33342. To measure oocyst size, images of midguts were acquired with a Leica DM 6000B fluorescent microscope and analyzed using ImageJ software: mCherry signal was binarized using the threshold function and oocyst size was measured with the 'analyze particles' function. Graphs and statistical analysis were performed with GraphPad Prism. Morphology of *Pb*HscB-KO oocysts was analyzed on a Leica DM5500 widefield fluorescence microscope. Midguts infected with *Pb*HscB-GFP expressing parasites were stained with 1 µg/mL Hoechst 33342 and 100 nM MitoView 650 (Biotium MitoView™ 650). Localization of *Pb*HscB-GFP signal was analyzed on a Leica TCS SP8 laser-scanning confocal microscope.

## Culture and *in vitro* infection of HeLa cells and quantification of salivary gland sporozoites

WT HeLa (European Cell Culture Collection) cells were cultivated in Minimum Essential Medium with Earle's salts (MEM EBS; Bioconcept, 1-31F01-I) supplemented with 10% FCS (Sigma-Aldrich), 100 U penicillin, 100 µg/mL streptomycin, and 2 mM L-glutamine (all from Bioconcept). Cells were kept at 37˚C with 5% $CO_2$ and split with Accutase (Innovative Cell Technologies). For parasite infections, salivary glands of infected mosquitoes were dissected and homogenized to release the sporozoites. Sporozoites were added to cells and incubated in normal growth medium. To determine number of salivary gland sporozoites, 10 µL of

sporozoite suspension was added in a Neubauer counting chamber and all 4 fields were counted. With the total volume used and the number of mosquitoes dissected, the average number of salivary gland sporozoites per mosquito was calculated.

## Automated microscopy

Infected HeLa cells were imaged live at several time points to measure number and size of parasites. Cells were imaged with an INCell Analyzer 2000 automated live cell imaging system (GE Healthcare Life Sciences). Analysis of images was carried out with INCell Developer Toolbox 1.10.0 software. Segmentation was done using "object" mode for the mCherry channel and objects smaller than 10 μm$^2$ and bigger than 1,200 μm$^2$ were excluded. Graphs and statistical analysis were done with GraphPad Prism.

## Indirect immunofluorescence assay

Infected cells were fixed at indicated time points with 4% paraformaldehyde in PBS 1X for 20 min at room temperature and protected from light. Cells were permeabilized for 30 min with -20°C 100% methanol. Unspecific antibody binding was blocked by incubating the cells for 20 min with 10% FCS in PBS. Cells were stained with primary antibodies diluted in 10% FCS in PBS for 1 h at room temperature. After washing with PBS, cells were incubated for another hour with fluorescently labelled secondary antibodies and DAPI (1 μg/mL). Primary antibodies used were rabbit anti-UIS4 antiserum (1:1000), rat anti-MSP1 antiserum (1:500) kindly gifted by Anthony Holder, mouse monoclonal anti-GFP (Roche 11814460001; 1:1000), mouse monoclonal anti-LC3 antibody (MBL M152-3; 1:1000) and rabbit anti-*Tg*Hsp70 antiserum (1:500) kindly gifted by Dominique Soldati. Secondary antibodies used were anti-rabbit Alexa488 (Invitrogen A- 11008; 1:2000), anti-rat Alexa647 (Invitrogen A-21247; 1:2000), anti-mouse Alexa488 (Invitrogen A- 11001, 1:2000) and anti-rabbit Cy5 (Dianova; 1:2000). Stained cells were imaged on a Leica TCS SP8 laser-scanning confocal microscope. Acquired images were processed with FIJI software. Quantification of LC3-labelled parasites was done on a Leica DM5500 epifluorescence microscope. Only UIS4-positive parasites were analyzed, and parasites were counted as LC3-positive when a clear association between LC3 and UIS4 was detected. To calculate the Pearson coefficient correlation (PCC) and the Mander overlap coefficient (MOC) only two channels (*Pb*HscB-GFP and *Tg*Hsp70) were retained, and the others were reduced using the 'arrange channels' function. The composite image was split into both channels and the region of interest (ROI) was defined in one of the channels (*Pb*HscB-GFP). Colocalization analysis were calculated by using the Coloc2 tool of the FIJI software.

## *In vivo* experiments

Salivary glands from infected mosquitoes were extracted and homogenized. Number of sporozoites per mL was determined with a Neubauer counting chamber. Sporozoites were diluted with PBS to inject 5,000 (Fig 2E) or 100,000 (Fig 4F) sporozoites in 150ul of PBS per mouse and injections were done via the tail vein of the mouse. In Fig 2E, 5,000 sporozoites per mouse (4 mice for WT and 4 mice for *Pb*HscB-KO) were injected intravenously into C57BL/6 mice per parasite line in two independent experiments. Mice were euthanized once parasitemia reached 2% at day 5 for mice infected with WT parasite line and between day 7 and 10 for mice infected with *Pb*HscB-KO parasite line. Represented is one experiment. In Fig 4F, C56BL/6 mice were infected with 100,000 sporozoites each (4 mice for WT, 3 mice for *Pb*HscB-KO, 4 mice for *Pb*Mei2-KO and 6 mice for *Pb*HscB-*Pb*Mei2-dKO). Mice were euthanized once parasitemia reached 2% (day 5 for mice infected with WT parasite line, day 7 for mice infected with *Pb*HscB-KO parasite line, between day 9 and 10 for *Pb*Mei2-KO parasite

line and at day 41 for *Pb*HscB-*Pb*Mei2-dKO parasite line). Parasitemia was measured daily using flow cytometry of the mCherry signal. 100 events and more per 1 Mio erythrocytes was considered as blood stage positive. Graphs were created with Microsoft Excel.

*In vivo* imaging: To detect liver stage parasites *in vivo* in Fig 4E, B6 Albino mice (B6(C)/Rj-Tyrc/c; JanvierLabs, France) were infected with 100,000 *Pb*HscB-*Pb*Mei2-dKO or the parental parasite line *Pb*mCherryhsp70-Lucef1a (*Pb*1868) named here as WT sporozoites by intravenous injection; salivary glands from mock infected mosquitoes served as negative control. At 44hpi full-body bioluminescence analysis was performed. Mice were anaesthetized using isofluorane and administered with 100ul of RediJect D-Luciferin (30mg/ml; Perkin Elmer, 770504) by intraperitoneal injection. Seven minutes after substrate administration the luciferase activity was measured for 3 minutes using the In Vivo Imaging System NightOWL LB983 (Berthold Technologies). Values are expressed as photons/sec and represent parasite load in the liver.

## Supporting information

**S1 Fig. Generation of transgenic parasite lines. (A)** Schematic representation of single crossover recombination of GFP-tagging vector at the PBANKA_0821000 locus. The C-terminus of PBANKA_0821000 (blue) was amplified and fused to a C-terminal GFP. Human dihydrofolate reductase (hdhfr) was used as a positive selection marker. Primers used to check for integration are marked as 1 and 2. **(B)** Agarose gel showing products of PCR done on genomic DNA extracted from WT and from transgenic *Pb*HscB-GFP parasites. Expected band size for the PCR: 2,100 bp. **(C)** Flow cytometry graph from WT and *Pb*HscB-GFP parasites showing mCherry-expressing parasites with a population of GFP-expressing parasites (green arrow). **(D)** Represents the Channel 1 (*Pb*HscB-GFP) and the Channel 2 (*Tg*Hsp70) at 48 hpi time-point (left panel) used to measure the Pearson correlation coefficient (PCC) and the Mander's overlap coefficient (MOC). The 2D intensity histogram of Channel 1 and Channel 2 (right panel) shows the ratio of intensities at the exact position in a density heat map (a high intensity is displayed in yellow, a low intensity in purple). The image at 48 hpi used correspond to the one presented in Fig 1B. **(E)** Schematic representation of double crossover recombination of the *Pb*HscB-KO vector (*Pb*GEM-239805) at the PBANKA_0821000 locus. Human dihydrofolate reductase (hdhfr) was used as a positive selection marker and yeast cytosine deaminase uracil phosphoribosyl transferase (yfcu) as a negative selection marker. The purple bar represents the gene-specific barcode. Blue bars represent *Pb*DHFR 3' UTRs. Primers used to check for integration are marked as 3–5. **(F)** Agarose gel showing PCR products done on genomic DNA of *Pb*HscB-KO parasites after limiting dilution. Expected band sizes are 2,450 bp for 3+4 (WT) and 2,130 bp for 3+5 (locus after integration of KO vector). Clone 1 was chosen for further analysis. Schematics created with BioRender.com.
(TIF)

**S2 Fig. *Pb*HscB-KO does not affect oocyst formation and development. (A)** Midguts of infected mosquitoes (n = 10, black: WT and red: *Pb*HscB-KO) were dissected and imaged on day 7, day 10, and day 14 post-feed. Images were acquired using a 5x objective and analyzed using ImageJ software: mCherry signal was binarized using the threshold function and the oocyst size was measured with the 'analyze particle's function. The shown result is a representative of three experiments and shows the size of 300 oocysts per parasite line as violin plot with individual oocyst sizes and medians with interquartile range. Kruskal-Wallis test did not result in a significant difference (ns) between WT and KO oocysts in none of the analyzed time points. ($p_{day7}$ = 0.9412, $p_{day10}$ = 0.4174, $p_{day14}$>0.9999). **(B)** Infected midguts were extracted on day 14 post-feed and DNA was stained with Hoechst 33342 (cyan). Cytosolic

mCherry is shown in red. Sporozoite formation in *Pb*HscB-KO oocysts is highlighted with a white arrow. Scale bars are 10 μm.
(TIF)

**S3 Fig. Generation of *Pb*HscB-*Pb*Mei2-dKO parasite line. (A)** Schematic representation of negative selection using 5-fluorocytosine (5-FC). *Pb*DHFR-3'UTRs can recombine, which removes the selection cassette. Marker-free parasites can be selected using 5-FC. The absence of the selection marker was checked with indicated primers 3–7. **(B)** Agarose gel showing PCR products done on genomic DNA of *Pb*1868 WT parasites, 5-FC selected parasites, and two 5-FC selected clones after limiting dilution. Expected PCR products are 2,450 bp for 3+4; 2,130 bp for 3+5; and 1,425 bp for 6+7. The absence of a selection marker was confirmed for clone 2, which was selected for the generation of the double KO line. **(C)** Schematic representation of double crossover recombination of the *Pb*Mei2-KO vector (*Pb*GEM-300555) at the PBANKA_1122300 locus. Human dihydrofolate reductase (hdhfr) was used as a positive selection marker and yeast cytosine deaminase uracil phosphoribosyl transferase (yfcu) as a negative selection marker. The purple bar represents the gene-specific barcode. Blue bars represent *Pb*DHFR 3' UTRs. Primers used to check for integration are marked as 8–10. **(D)** Agarose gel showing PCR products done on genomic DNA of *Pb*1868 WT and *Pb*HscB-*Pb*Mei2-dKO parasites after transfection and after limiting dilution. Expected band sizes are 2,711 bp for 8+10 (WT) and 2,243 bp for 8+9 (locus after integration of KO vector). Clone 1 was chosen for further analysis. Schematics created with BioRender.com.
(TIF)

**S1 Raw image. Raw images of agarose gel electrophoresis.**
(PDF)

## Acknowledgments

We are grateful to Chris Janse and Blandine Franke-Fayard for providing the *Pb*Mei2-KO parasite line; to Anthony Holder for the antibody anti-MSP1; to Dominique Soldati for the anti-*Tg*Hsp70 antiserum. We thank the MIC (Microscopy Imaging Center) in Bern for providing excellent imaging facilities and expertise.

## Author Contributions

**Conceptualization:** Volker Heussler, Magali Roques.

**Formal analysis:** Melanie Schmid, Raphael Beyeler, Reto Caldelari, Magali Roques.

**Funding acquisition:** Volker Heussler.

**Investigation:** Melanie Schmid, Raphael Beyeler, Ruth Rehmann, Magali Roques.

**Methodology:** Melanie Schmid, Raphael Beyeler, Reto Caldelari, Ruth Rehmann.

**Project administration:** Volker Heussler, Magali Roques.

**Supervision:** Volker Heussler, Magali Roques.

**Validation:** Melanie Schmid, Raphael Beyeler, Reto Caldelari, Volker Heussler, Magali Roques.

**Visualization:** Melanie Schmid, Raphael Beyeler, Reto Caldelari.

**Writing – original draft:** Raphael Beyeler, Volker Heussler, Magali Roques.

**Writing – review & editing:** Melanie Schmid, Reto Caldelari, Volker Heussler, Magali Roques.

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
