## [Decision Letter · Decision Letter 0]

16 Sep 2024

PONE-D-24-28684Generation of a genetically double-attenuated Plasmodium berghei parasite that fully arrests growth during late liver stage developmentPLOS ONE

Dear Dr. Roques,

Thank you for submitting your manuscript to PLOS ONE. After careful consideration, we feel that it has merit but does not fully meet PLOS ONE’s publication criteria as it currently stands. Therefore, we invite you to submit a revised version of the manuscript that addresses the points raised during the review process.

We look forward to receiving your revised manuscript.

Kind regards,

Shahin Tajeri, D.V.M. Ph.D.

Academic Editor

PLOS ONE

Journal requirements: 1. When submitting your revision, we need you to address these additional requirements. Please ensure that your manuscript meets PLOS ONE's style requirements, including those for file naming. The PLOS ONE style templates can be found at https://journals.plos.org/plosone/s/file?id=wjVg/PLOSOne_formatting_sample_main_body.pdf and https://journals.plos.org/plosone/s/file?id=ba62/PLOSOne_formatting_sample_title_authors_affiliations.pdf. 2. PLOS ONE now requires that authors provide the original uncropped and unadjusted images underlying all blot or gel results reported in a submission’s figures or Supporting Information files. This policy and the journal’s other requirements for blot/gel reporting and figure preparation are described in detail at https://journals.plos.org/plosone/s/figures#loc-blot-and-gel-reporting-requirements and https://journals.plos.org/plosone/s/figures#loc-preparing-figures-from-image-files. When you submit your revised manuscript, please ensure that your figures adhere fully to these guidelines and provide the original underlying images for all blot or gel data reported in your submission. See the following link for instructions on providing the original image data: https://journals.plos.org/plosone/s/figures#loc-original-images-for-blots-and-gels.   In your cover letter, please note whether your blot/gel image data are in Supporting Information or posted at a public data repository, provide the repository URL if relevant, and provide specific details as to which raw blot/gel images, if any, are not available. Email us at plosone@plos.org if you have any questions. 3. Your ethics statement should only appear in the Methods section of your manuscript. If your ethics statement is written in any section besides the Methods, please move it to the Methods section and delete it from any other section. Please ensure that your ethics statement is included in your manuscript, as the ethics statement entered into the online submission form will not be published alongside your manuscript. 

Additional Editor Comments:

Dear Dr. Roques,

I apologize for the extended review process. At this time of year, many reviewers were unable to participate, making it challenging to find suitable experts. However, I have now received two independent reviews. Your manuscript is close to being accepted, pending your responses to the minor comments from the reviewers.

Best regards,

Shahin Tajeri

Reviewers' comments:

Reviewer's Responses to Questions

**Comments to the Author**

1. Is the manuscript technically sound, and do the data support the conclusions?

Reviewer #1: Yes

Reviewer #2: Yes

2. Has the statistical analysis been performed appropriately and rigorously? 

Reviewer #1: Yes

Reviewer #2: Yes

3. Have the authors made all data underlying the findings in their manuscript fully available?

Reviewer #1: Yes

Reviewer #2: Yes

4. Is the manuscript presented in an intelligible fashion and written in standard English?

Reviewer #1: Yes

Reviewer #2: Yes

5. Review Comments to the Author

Reviewer #1: This is a nice and straight forward paper describing the generation of a new double gene deletion mutant in Plasmodium berghei as a model for attenuated parasites. Clearly further tests are needed as the authors state in the discussion but as a first report the paper is technically sound, provides a new and interesting tool and novel insights. The following are suggestions for slight improvement:

What is not clear to me is if MSP1 is expressed in mei2/linup KOs. This could be interesting and a possible point to discuss: If no MSP1 is expressed likely it’s not very late-stage development and no antibodies are made to protect from a blood stage infection. Hence, a breakthrough infection will likely lead to full blown infection.

Lines 41/42: new data from Sinnis Lab suggests that Anopheles transmits 500-1000 SPZ, please modify (PMID: 38272943)

43: also transmigrate through skin cells (PMID: 18312843)

56: new data suggest under 20% efficacy in implementation studies

94/95 This paper PMID: 27241521 reports the lowest rate of breakthrough in a GAP to my knowledge and should be mentioned if low number of mice are used to test for breakthroughs – could also be discussed in the paragraph around line 460

482: please change ‘prove’ into ‘test’

Reviewer #2: Among the different vaccine approaches against malaria, immunization with whole attenuated sporozoites is considered an interesting strategy since it allows to induce sterilizing immunity based on immune mechanisms directed against the hepatic stages of malaria, the initial phase of parasite multiplication in the vertebrate host.

While the first vaccination tests dating back to the 1970s used sporozoites attenuated by irradiation, current strategies focus on using either non-attenuated parasites with drug cover (« chemically attenuated »), or developping genetically attenuated parasites. This latter solution, requiring less medical monitoring constraints, appears most promising provided the attenuation is robust enough to prevent any breakthrough infections when immunizing with live sporozoites.

In their manuscript entitled " Generation of a genetically double-attenuated Plasmodium berghei parasite that fully arrests growth during late liver stage development" Schmid et al. first investigate the localization of HscB in the rodent parasite Plasmodium berghei within mosquitoes and at the liver stage, using a P. berghei parasite that they genetically engineered to express the GFP reporter gene. They documented the localization of PbHscB at the mitochondrial level in oocysts within mosquitoes, and in exo erythrocytic forms in culture.

The authors demonstrated that deleting PbHscB has no effect on the mosquito stage, thus allowing sporozoite production. However, PbHscB knockout parasites exhibit significant growth and maturation defects at the liver stage, resulting in a prolonged patency period when injected into mice, though infection eventually occurs.

The authors then engineered a second parasite with double deletion for both the HscB gene and Pbmei2, previously shown to be essential for the completion of Plasmodium liver stages, although with a significant risk of breakthrough. This double knockout parasite develops normally in mosquitoes, producing infectious sporozoites, but is completely arrested at the liver stage and cannot initiate a blood-stage infection, even when a massive dose of sporozoites is injected into mice.

The manuscript is technically sound with data that generally support the conclusions, including appropriate statistical analysis and biological replicates. All data underlying the conclusions are available in the main and supplementary figures.

However, a few minor points could benefit from additional clarification or analysis:

The authors refer to a phenotypic screening in reference 34 for their choice to study PbHscB in detail. It is unclear where PbHscB is specifically discussed in that reference. The authors should verify this citation.

In Figure 1, the authors use mitochondrial markers to conclude colocalization of PbHscB at the mitochondrial level based on confocal microscopy. Merged images alone are often insufficient for confirming colocalization. Utilizing image analysis tools that measure pixel intensity or calculate correlation coefficients, such as the Pearson coefficient, would strengthen this conclusion.

In Figure 1B, Hela cells are used to study liver stage features of P. berghei, but these cells differ from natural host cells. Including images of PbHscB liver stages in murine hepatocytes or liver slices from infected mice, or at least in an hepatic origin cell line, would provide more relevant data.

Figure 4B assesses the sporozoite production of the PbHscB-Pbmei2-dKO clone, but only four data points are shown. Clarifying whether these points represent averages from multiple experiments or sporozoite counts from individual mosquitoes would be helpful.

An additional experiment to demonstrate the immunogenicity of this double knockout parasite, such as vaccinating mice followed by a challenge with infectious sporozoites, could further validate the study, though protection is likely, given the late arrest of the parasite.

The discussion section, while informative, is somewhat lengthy and contains redundancies. The "experimental procedure" section also could benefit from clearer and more concise language, particularly regarding the redundancy between "Animal experiments conducted at the University of Bern" and "in vivo experiments."

In conclusion, addressing these points will enhance the manuscript. Developing robustly genetically attenuated Plasmodium parasites to minimize breakthrough risks is crucial for advancing malaria vaccines. Blocking parasite development at a late liver stage to ensure broad antigenic presentation and optimize the immune response is a significant challenge. The strategy of targeting different pathways for attenuation, as demonstrated in this study, is a promising approach to prevent breakthrough infections.

6. PLOS authors have the option to publish the peer review history of their article (what does this mean?). If published, this will include your full peer review and any attached files.

Reviewer #1: No

Reviewer #2: No

---

## [Author Response · Author response to Decision Letter 0]

20 Nov 2024

Responses to the Reviewers:

We strongly appreciate the Reviewers’s comments to improve and clarify our work. We made the changes accordingly and responded to all the requests in blue below the comments. 

REVIEWER COMMENTS:

Reviewer #1: This is a nice and straight forward paper describing the generation of a new double gene deletion mutant in Plasmodium berghei as a model for attenuated parasites. Clearly further tests are needed as the authors state in the discussion but as a first report the paper is technically sound, provides a new and interesting tool and novel insights. The following are suggestions for slight improvement:

What is not clear to me is if MSP1 is expressed in mei2/linup KOs. This could be interesting and a possible point to discuss: If no MSP1 is expressed likely it’s not very late-stage development and no antibodies are made to protect from a blood stage infection. Hence, a breakthrough infection will likely lead to full blown infection.

Thank you for your insightful comment. Indeed, in our study, there was no detectable MSP1 expression in the PbHscB-PbMei2-double knockout (dKO) parasites. This observation aligns with the expectation of limited cross-reaction and suggests that the parasites do not reach very late-stage liver development. The absence of MSP1 expression is significant because it implies that antibodies targeting this protein, which could contribute to protection against blood-stage infection, are not generated.

This raises an interesting point for discussion: if MSP1 is not expressed in double knockouts, the parasites may not progress to the stage where protective antigens are sufficiently presented to elicit an immune response. Consequently, should a breakthrough infection occur, it could likely result in a complete blood-stage infection due to the absence of prior protective immunity. We will highlight this consideration in the revised discussion to emphasize the implications of MSP1 absence and its potential impact on immune protection and breakthrough infections.

Lines 41/42: new data from Sinnis Lab suggests that Anopheles transmits 500-1000 SPZ, please modify (PMID: 38272943)

Reference added and changes made.

43: also transmigrate through skin cells (PMID: 18312843)

Reference added and changes made.

56: new data suggest under 20% efficacy in implementation studies

Change made.

94/95 This paper PMID: 27241521 reports the lowest rate of breakthrough in a GAP to my knowledge and should be mentioned if low number of mice are used to test for breakthroughs – could also be discussed in the paragraph around line 460

We appreciate the reviewer’s suggestion regarding the inclusion of the study reported in Kumar et al, 2016. This paper will be cited to highlight an example of a GAP model with a low breakthrough rate, which is relevant to our discussion on the potential limitations of using a small number of mice for breakthrough testing. We will incorporate this reference in our revised manuscript to provide context and comparison with our findings. 

482: please change ‘prove’ into ‘test’

Change made.

Reviewer #2: Among the different vaccine approaches against malaria, immunization with whole attenuated sporozoites is considered an interesting strategy since it allows to induce sterilizing immunity based on immune mechanisms directed against the hepatic stages of malaria, the initial phase of parasite multiplication in the vertebrate host.

While the first vaccination tests dating back to the 1970s used sporozoites attenuated by irradiation, current strategies focus on using either non-attenuated parasites with drug cover (« chemically attenuated »), or developping genetically attenuated parasites. This latter solution, requiring less medical monitoring constraints, appears most promising provided the attenuation is robust enough to prevent any breakthrough infections when immunizing with live sporozoites.

In their manuscript entitled " Generation of a genetically double-attenuated Plasmodium berghei parasite that fully arrests growth during late liver stage development" Schmid et al. first investigate the localization of HscB in the rodent parasite Plasmodium berghei within mosquitoes and at the liver stage, using a P. berghei parasite that they genetically engineered to express the GFP reporter gene. They documented the localization of PbHscB at the mitochondrial level in oocysts within mosquitoes, and in exo erythrocytic forms in culture.

The authors demonstrated that deleting PbHscB has no effect on the mosquito stage, thus allowing sporozoite production. However, PbHscB knockout parasites exhibit significant growth and maturation defects at the liver stage, resulting in a prolonged patency period when injected into mice, though infection eventually occurs.

The authors then engineered a second parasite with double deletion for both the HscB gene and Pbmei2, previously shown to be essential for the completion of Plasmodium liver stages, although with a significant risk of breakthrough. This double knockout parasite develops normally in mosquitoes, producing infectious sporozoites, but is completely arrested at the liver stage and cannot initiate a blood-stage infection, even when a massive dose of sporozoites is injected into mice.

The manuscript is technically sound with data that generally support the conclusions, including appropriate statistical analysis and biological replicates. All data underlying the conclusions are available in the main and supplementary figures.

However, a few minor points could benefit from additional clarification or analysis:

The authors refer to a phenotypic screening in reference 34 for their choice to study PbHscB in detail. It is unclear where PbHscB is specifically discussed in that reference. The authors should verify this citation.

Thank you for pointing out the need to verify the citation. The reference to Stanway et al., 2019 (now cited as reference 37, PMID: 31730853) is indeed accurate. This publication by our group and collaborators details the identification of key genes involved in Plasmodium berghei liver stage development through a high-throughput phenotypic screen. Although the main focus was on metabolic pathways such as fatty acid and amino sugar biosynthesis, the barcode abundance data for structural gene deletions and others like the PbHscB-knockout (PBANKA_0821000) is included in Table S2 of the study by Stanway et al. 

Specifically, the PbHscB-KO parasite showed a 19-fold reduction (Log2FC = -4.23) in barcode abundance at the salivary gland sporozoite (SG) to blood-stage transition (B2), indicating significant impairment during liver stage development. This marked reduction drew our attention to HscB as a candidate for further investigation. We have now clarified this in the manuscript.

 Please find below the part of table S2 in Stanway et al recapitulating on the HscB KO barcoding throughout the entire life cycle of P. berghei:

 B1-MG data, normalized

 MG-SG data, normalized

 SG-B2 data, normalized, BS fitness-corrected

 Blood stage fitness data (Bushell et al., 2017)

 Log2-FC SD Power Log2-FC SD Power Log2-FC SD Power Relative growth Phenotype 

PBANKA_0821000 -1.42 0.24 no power -1.12 0.33 no power -4.23 0.60 reduced 0.82 Slow

In Figure 1, the authors use mitochondrial markers to conclude colocalization of PbHscB at the mitochondrial level based on confocal microscopy. Merged images alone are often insufficient for confirming colocalization. Utilizing image analysis tools that measure pixel intensity or calculate correlation coefficients, such as the Pearson coefficient, would strengthen this conclusion.

We appreciate the reviewer’s insightful comment regarding the need for quantitative confirmation of colocalization. In response, we have now included additional analysis in Supplementary Figure 1D, which features the Pearson’s correlation coefficient and Manders' overlap coefficients for P. berghei liver stage parasites at 48 hours post-infection, based on the image in Figure 1B. The analysis includes channel 1 (PbHscB-GFP) and channel 2 (TgHsp70) on the left, alongside a 2D intensity histogram of both channels on the right.

The measured Pearson’s R value was 0.44, with Manders' coefficients tM1 at 0.853 and tM2 at 0.894, indicating a moderate level of overlap between the two signals. We have also updated the figure panel numbers in the main text, revised the legend for Supplementary Figure 1, and included details of these calculations in the Materials and Methods section.

These data are now provided in the revised Supplementary Figure 1, complete with an updated figure legend (see below).

S1 Fig: Generation of transgenic parasite lines. (A) Schematic representation of single crossover recombination of GFP-tagging vector at the PBANKA_0821000 locus. The C-terminus of PBANKA_0821000 (blue) was amplified and fused to a C-terminal GFP. Human dihydrofolate reductase (hdhfr) was used as a positive selection marker. Primers used to check for integration are marked as 1 and 2. (B) Agarose gel showing products of PCR done on genomic DNA extracted from WT and from transgenic PbHscB-GFP parasites. Expected band size for the PCR: 2,100 bp. (C) Flow cytometry graph from WT and PbHscB-GFP parasites showing mCherry-expressing parasites with a population of GFP-expressing parasites (green arrow). (D) Represents the Channel 1 (PbHscB-GFP) and the Channel 2 (TgHsp70) at 48 hpi timepoint (left panel) used to measure the Pearson correlation coefficient (PCC) and the Mander’s overlap coefficient (MOC). The 2D intensity histogram of Channel 1 and Channel 2 (right panel) shows the ratio of intensities at the exact position in a density heat map (a high intensity is displayed in yellow, a low intensity in purple). The image at 48 hpi used correspond to the one presented in Fig 1B. (E) Schematic representation of double crossover recombination of the PbHscB-KO vector (PbGEM-239805) at the PBANKA_0821000 locus. Human dihydrofolate reductase (hdhfr) was used as a positive selection marker and yeast cytosine deaminase uracil phosphoribosyl transferase (yfcu) as a negative selection marker. The purple bar represents the gene-specific barcode. Blue bars represent PbDHFR 3’ UTRs. Primers used to check for integration are marked as 3 – 5. (F) Agarose gel showing PCR products done on genomic DNA of PbHscB-KO parasites after limiting dilution. Expected band sizes are 2,450 bp for 3+4 (WT) and 2,130 bp for 3+5 (locus after integration of KO vector). Clone 1 was chosen for further analysis. Schematics created with BioRender.com.

In Figure 1B, Hela cells are used to study liver stage features of P. berghei, but these cells differ from natural host cells. Including images of PbHscB liver stages in murine hepatocytes or liver slices from infected mice, or at least in an hepatic origin cell line, would provide more relevant data. 

We appreciate the Reviewer’s concerns regarding the use of HeLa cells for studying liver stage features of P. berghei. It is important to note that the suitability of HeLa cells for P. berghei liver stage studies has been well-documented in the literature. Multiple studies have demonstrated that HeLa cells are permissive for P. berghei infection and serve as a reliable model for initial in vitro assessments of parasite development, particularly for high-throughput or mechanistic investigations. In addition to the in vitro data, we have conducted comprehensive in vivo experiments in murine models, which demonstrate solid attenuation of the double knockout parasite as attested in Figure 4. We have now verified the PbHscB-PbMei2-dKO salivary glands sporozoites’ ability to reach and invade mouse hepatocytes in vivo and added the new data in Figure 4E. For this, we infected B6 Albino mice with 100,000 WT (positive control) or PbHscB-PbMei2-dKO salivary glands sporozoites and performed a full-body bioluminescence analysis at 44 hours post-sporozoite infection. The mock (non-infected mosquito salivary glands) was used as a negative control and the number of mosquitoes dissected was the same as for WT. Although the luminescence signal from the double-attenuated parasites is understandably low due to a tenfold size reduction compared to the control (as shown in Figure 4A), we clearly demonstrate that PbHscB-PbMei2-dKO parasites can reach and invade mouse hepatocytes in vivo. This new data has been incorporated into Figure 4E, the figure legend, main text, and the Materials and Methods section: 

Fig 4: PbMei2-KO and PbHscB-PbMei2-dKO parasites are arrested in the liver stage. (A) HeLa cells were infected with either WT or PbHscB-KO sporozoites and imaged at indicated time points using automated microscopy. The shown result is a representative of three experiments and shows parasite size of 500 parasites measured based on cytosolic mCherry signal expressed. The significance of the results was determined with Kruskal-Wallis test; (****: p<0.0001, ns: p24h1868-Mei2KO=0.9528, p48h1868-Mei2KO=0.076, p56h1868-Mei2KO=0.0674). (B) The number of parasites was determined at 6 hpi and 48 hpi by automated microscopy and the number at 6 hpi was considered as 100 %. Significance was calculated by Kruskal-Wallis test, with Dunn’s multiple comparison test (*: p=0.0276; ns: p1868-Mei2KO>0.9999, p1868-dKO=0.4231). Each dot represents an infection experiment. (C) HeLa cells infected with WT or PbMei2-KO parasites were fixed at 56 hpi and stained with DAPI (DNA, blue), anti-TgHsp70 antibodies (mitochondrion, green), and anti-MSP1 antibodies (parasite plasma membrane, magenta). Cytosolic mCherry signal is shown in red. Scale bars are 10 µm. (D) HeLa cells infected with WT or PbMei2-KO parasites were fixed at 56 hpi and stained with DAPI (DNA, blue) and antibodies against UIS4 (PVM, magenta). Cytosolic mCherry is shown in red. Scale bars are 10 µm. (E) Albino mice (B6(C)/Rj-Tyrc/c) injected with 100,000 sporozoites from mosquito salivary glands infected with either PbHscB-PbMei2-dKO or WT parasites were assessed for luciferase activity via luminescence measurement. Salivary glands from uninfected P. berghei mosquitoes served as a negative control (Mock). At 44 hpi mice were anaesthetized, injected with luciferin and parasite load in the liver was analyzed using an In Vivo Imaging System (IVIS). The graph displays the mean total photon counts per second, with each dot representing an individual mouse (N=3 per group). Statistical significance was determined using an unpaired t-test. (F) C56BL/6 mice were infected with 100,000 sporozoites each (4 mice for WT, 3 mice for PbHscB-KO, 4 mice for PbMei2-KO and 6 mice for PbHscB-PbMei2-dKO). Parasitemia was monitored daily by flow cytometry and a parasitemia of 0.01% (100 parasites per 1 million erythrocytes) was set as threshold. In a pilot study, it was already confirmed that infection of mice with 100,000 PbMei2-KO sporozoites resulted in breakthrough infections. 

Figure 4B assesses the sporozoite production of the PbHscB-Pbmei2-dKO clone, but only four data points are shown. Clarifying whether these points represent averages from multiple experiments or sporozoite counts from individual mosquitoes would be helpful.

We thank the Reviewer for pointing out the missing information in the Figure 3 legend. Each data point shown in the graph of Figure 3B represents the mean of salivary gland sporozoites number from 10 mosquitoes for one experiment, meaning that four independent experiments are represented in the graph in Fig 3B. See below the new figure legend:

Fig 3: PbMei2-KO and PbHscB-PbMei2-dKO do not affect oocyst formation and sporozoite production. (A) Midguts of infected mosquitoes were dissected (n=10) and imaged on day 7 and day 14 post-feeding. Images were acquired using a 5x objective and analyzed using ImageJ software: mCherry signal was binarized using the threshold function and oocyst size was measured with the analyze particles function. The presented result is a representative of three experiments and shows 240 oocysts per parasite line as violin plot with individual oocyst sizes and medians with interquartile ranges. Kruskal-Wallis test did not result in a significant difference (ns) between WT and modified parasite line oocysts in neither at day 7 nor at

---

## [Decision Letter · Decision Letter 1]

8 Dec 2024

Generation of a genetically double-attenuated Plasmodium berghei parasite that fully arrests growth during late liver stage development

PONE-D-24-28684R1

Dear Dr. Roques,

We’re pleased to inform you that your manuscript has been judged scientifically suitable for publication and will be formally accepted for publication once it meets all outstanding technical requirements.

Kind regards,

Shahin Tajeri, D.V.M. Ph.D.

Academic Editor

PLOS ONE

Additional Editor Comments (optional):

Reviewers' comments:

Reviewer's Responses to Questions

**Comments to the Author**

1. If the authors have adequately addressed your comments raised in a previous round of review and you feel that this manuscript is now acceptable for publication, you may indicate that here to bypass the “Comments to the Author” section, enter your conflict of interest statement in the “Confidential to Editor” section, and submit your "Accept" recommendation.

Reviewer #1: All comments have been addressed

Reviewer #2: All comments have been addressed

2. Is the manuscript technically sound, and do the data support the conclusions?

Reviewer #1: Yes

Reviewer #2: Yes

3. Has the statistical analysis been performed appropriately and rigorously? 

Reviewer #1: Yes

Reviewer #2: Yes

4. Have the authors made all data underlying the findings in their manuscript fully available?

Reviewer #1: Yes

Reviewer #2: Yes

5. Is the manuscript presented in an intelligible fashion and written in standard English?

Reviewer #1: Yes

Reviewer #2: Yes

6. Review Comments to the Author

Reviewer #1: The authors have addressed all my questions adequately by adding text and references where appropriate

Reviewer #2: The authors have adequately addressed my comments and clarified the points I raised when reading the first version of the manuscript. The additional experiments and data that have now been added strengthen the conclusions, and the corrections made in the text have significantly improved the reading of the manuscript. This manuscript now appears to me to be acceptable for publication.

7. PLOS authors have the option to publish the peer review history of their article (what does this mean?). If published, this will include your full peer review and any attached files.

Reviewer #1: No

Reviewer #2: No

---

## [Editor Report · Acceptance letter]

16 Dec 2024

PONE-D-24-28684R1 

PLOS ONE

Dear Dr. Roques, 

I'm pleased to inform you that your manuscript has been deemed suitable for publication in PLOS ONE. Congratulations! Your manuscript is now being handed over to our production team.

Kind regards, 

on behalf of

Dr. Shahin Tajeri 

Academic Editor

PLOS ONE